

# Disrupted Flow Memory and Synchrony in the Mekong River under Dam Regulation and Climate Change: Implications for Tonle Sap Reverse Flow

**Khosro Morovati[1], Hongling Zhao[1], Fuqiang Tian[1]**

[1]Department of Hydraulic Engineering & State Key Laboratory of Hydro-science and Engineering, Tsinghua University, Beijing, 100084, China

*Correspondence to*: Fuqiang Tian (tianfq@mail.tsinghua.edu.cn)

**Abstract.** Dam construction and climate change have profoundly disrupted the hydrological dynamics of the Mekong River and its floodplain–lake system. This study provides an integrated, multi-scale assessment of flow regime alteration across the

Mekong mainstream and its coupling with Tonle Sap Lake for three periods: pre-dam (1976–1991), transition (1992–2009), and post-dam (2010–2024). Using daily and sub-daily data from eight stations, we quantify changes in long-term flow memory, short-term variability, and flow amplitude. These analyses are complemented by a hydrodynamic response-time model that simulates how upstream regime shifts reshape the Tonle Sap–Mekong flow exchange.

We show that dam regulation and climate change have fragmented both spatial synchrony and temporal persistence,

disrupting the Mekong's behavior as a coherent hydrological continuum. Upstream stations, particularly Chiang Saen, exhibit a 33.3% increase in minimum flows and a 40% decrease in maximum flows during the post-dam period compared to the pre-dam period, alongside higher flow memory (+0.13), reflecting flow smoothing by reservoir operations. Downstream variability remains more pronounced due to the influence of monsoonal tributaries. At the most downstream station–Kratie, a key control point for Tonle Sap inflow–maximum flows decreased by 9.4%, while minimum flows increased by 117% in the

post-dam period relative to the pre-dam baseline, also delaying the seasonal timing of peak discharge by two weeks.

Critically, these regime shifts–driven by the compounding impacts of dam regulations and climate change, together with observed riverbed lowering from sand mining, caused the discharge threshold for initiating reverse flow to rise significantly: the median onset flow increased from ~3,000 m³ s⁻¹ (pre-dam) to ~7,000 m³ s⁻¹ (post-dam), marking a >130% increase. Collectively, these alterations have shortened the reverse flow period by 24 days compared to the historical baseline. Our

findings demonstrate that cascading dam operations across the mainstream and tributaries, amplified by climate change and other anthropogenic stressors, have triggered multi-scalar hydrological fragmentation in one of the world's most ecologically productive river–lake systems. Preserving the Mekong's natural flood pulse and its interaction with Tonle Sap lake will require basin-wide hydrological monitoring and transboundary governance frameworks that account not only for volume but also for flow timing, variability, and ecological function.



## 1 Introduction


In riverine ecosystems, the flow regime–comprising the magnitude, timing, frequency, duration, and rate of change of discharge–is a foundational driver of ecological structure and biogeochemical functioning (Poff et al., 1997; Binh et al., 2020). Alterations to this regime, driven by climate change and intensified human interventions such as dam construction, have emerged as critical concerns within the hydrological and ecological sciences (Lao et al., 2022; Morovati et al., 2024a).

Since the 1990s, global river management has shifted beyond managing total runoff volumes toward safeguarding the integrity of natural flow dynamics (Li et al., 2017). Maintaining or restoring ecologically functional flow regimes has thus become a shared priority for engineers, policymakers, and ecologists seeking to balance infrastructure development with ecosystem resilience (Yin et al., 2011; Arthington et al., 2018).

The Mekong River–one of the world's most biodiverse and socioeconomically critical transboundary rivers (Try et al., 2022)

– has undergone profound hydrological transformations due to the rapid expansion of hydropower infrastructure (Soukhaphon et al., 2021; Yun et al., 2024). Particularly since 2010, large–scale dam construction in the upper Mekong (Lancang) and lower Mekong tributaries has disrupted the river's natural monsoonal flood regime, fundamentally altering the hydrological connectivity that supports downstream floodplain and delta systems (Hecht et al., 2019; Zhang et al., 2023; Morovati et al., 2024a). From 2010 to 2015 alone, dam operations accounted for approximately 62% of total streamflow

change in the basin (Li et al., 2017). These impacts have been especially pronounced during the dry season: Räsänen et al. (2017) reported flow increases at Chiang Saen ranging from 121 to187% in March, and from 32 to 46% at Kratie. Lu and Chua (2021) found a 98% increase in monthly discharge at Chiang Saen during the dry months. Concurrently, wet-season flows have declined substantially (Lu et al., 2014), undermining the amplitude and timing of flood pulses that sustain floodplain ecosystems. Nguyen et al. (2025) documented a 73.7% increase in dry-season flows at Chiang Saen between 2000

and 2019, underscoring the dominant role of dam-induced regulation.

While such studies have advanced our understanding of hydrological alterations, limited empirical research has examined how regulation disrupts intra-annual flow memory – eroding the natural seasonal recurrence of discharge – or fragments synchrony across stations, thereby decoupling linked hydrological and ecological responses (Poff et al., 2007). In parallel, the effects of such disruptions on threshold-sensitive processes – such as the annual reverse flow from the Mekong into

Tonle Sap Lake, a globally unique hydrological phenomenon essential to Cambodian fisheries, agriculture, and biodiversity (Yoshida et al., 2020; Morovati et al., 2024b) – remain underexplored. Tonle Sap Lake functions as a seasonal water storage system, with reverse flows during the monsoon expanding the lake by up to sixfold (Try et al., 2020; Do et al., 2025), delivering nutrient-rich sediment and sustaining livelihoods across Cambodia (Arias et al., 2014; Morovati et al., 2023). This phenomenon is highly sensitive to discharge magnitude and timing at the Mekong-Tonle Sap confluence, with positive water

level gradients between the Mekong and the lake acting as the physical trigger (Hackney et al., 2025). However, dam-induced flow dampening (Dang et al., 2022), upstream climate change (e.g., shifts in precipitation patterns) (Zhang et al., 2023; Morovati et al., 2024), and extensive sand mining near the confluence have progressively delayed the onset of this



gradient, reducing the volume and duration of reverse flow (Hackney et al., 2025). Despite growing concern, no study has yet quantified how thresholds of flow needed to initiate or terminate reverse flow have evolved across decades, nor how dam regulation, climate change, and channel degradation jointly shape this regime shift (e.g., Chen et al., 2021; Chua et al., 2022; Morovati et al., 2023).

A wide range of approaches – including statistical indicators, hydrodynamic simulations, and hydrological models – have been employed to assess flow regime alterations in the Mekong River basin (Delgado et al., 2010; Lauri et al., 2012; Piman et al., 2013; Cochrane et al., 2014; Li et al., 2017; Lu and Chua, 2021; Wang et al., 2021; Yun et al., 2020). Among these, the Indicators of Hydrologic Alteration (IHA) framework remains one of the most widely used tools for evaluating site-specific changes in hydrologic parameters (Li et al., 2017; Binh et al., 2020). However, IHA's predominantly univariate and station-based nature limits its ability to capture system–wide dynamics, such as spatial coherence, inter-station synchrony, or long-term predictability. This limitation is particularly critical in the Mekong basin, where pronounced spatial and temporal variability in precipitation (Pokhrel et al., 2018; Zhang et al., 2023), combined with the presence of over 300 operational dams–and, according to some sources, more than 500 (Ang et al., 2024)–has resulted in extensive flow regulation. To address these gaps, alternative metrics have been proposed. The Richards–Baker (R–B) Flashiness Index, for example, captures short-term variability by quantifying how rapidly flow rises and falls (Gannon et al., 2022), while the Hurst exponent provides a measure of long-range dependence or memory in discharge time series.

This study addresses these gaps through a multi-decadal, multi-indicator analysis of flow regime shifts at eight mainstream stations along the Mekong River from 1976 to 2024. We quantify changes in daily discharge variability, flashiness, and memory across three hydrological periods: pre-dam (1976–1991), transition (1992–2009), and post-dam (2010–2024), introducing the concepts of disrupted flow memory and fragmented synchrony to describe the breakdown in spatiotemporal coherence. In addition to these indicators, we examine annual discharge extremes, including maximum and minimum daily flows and their associated timing. We further link these altered flow characteristics to the onset, termination, and duration thresholds of Tonle Sap's reverse flow by applying a hydrodynamically informed, travel-time-adjusted alignment of Kratie discharge data with lake response. This approach enables the empirical derivation of reverse flow thresholds and reveals significant temporal shifts in both their magnitude and timing. Our findings highlight the intensifying influence of recent dam developments, together with the impacts of climate change, and provide updated, system-wide insights for hydrological management within the frameworks of the Mekong River Commission (MRC) and the Lancang–Mekong Cooperation (LMC).

## 2 Materials and methods

### 2.1 Study area

The Mekong River, one of the world's most prominent transboundary river systems, extends over 4,800 km from the glaciated Tibetan Plateau through six countries – China, Myanmar, Laos, Thailand, Cambodia, and Vietnam – before



discharging into the sea (Duc et al., 2020). Over the past three decades, hydrological dynamics in the basin have been reshaped by the combined pressures of climate change and rapid hydropower expansion. Shifts in precipitation regimes and temperature variability have interacted with extensive dam construction on both the mainstream and tributaries (Ang et al., 2024), amplifying alterations to the river's seasonal flow regime. Eleven large dams are now located on the upper Mekong (Lancang) in China, with a collective storage capacity exceeding 43,000 MCM. Among them, the Xiaowan (14,645 MCM,

completed in 2010) and Nuozhadu (21,749 MCM, completed in 2014) reservoirs dominate in both scale and operational influence (Zhang et al., 2023).

In the lower Mekong, mainstream dam development began more recently, marked by the commissioning of the Xayaburi Dam in 2019 and the Don Sahong Dam in 2020, together contributing around 1,300 MCM of storage (Figure 1). Although many of the tributary dams are small (<5 km³), their cumulative storage, asynchronous operational strategies, and often

uncoordinated management (Lyu et al., 2024; Morovati et al., 2024b) – even within a single jurisdiction – have led to profound disruptions in the basin's natural flow regime. These hydrological shifts are further intensified by the region's pronounced spatial and temporal variability in precipitation, including extreme monsoonal rainfall patterns typical of the lower Mekong's tropical climate (Pokhrel et al., 2018; Zhang et al., 2023).

Downstream, the Mekong River interacts seasonally with the Tonle Sap Lake via the Tonle Sap River. During the wet

season, strong mainstream flows cause a hydraulic gradient reversal, pushing water back into the lake and expanding its surface area dramatically – from approximately 2,500 km² to over 13,000 km² (Kummu et al., 2014). This seasonal flood pulse is vital for the delivery of sediment and nutrient-rich water, supporting one of the world's most productive freshwater fisheries and sustaining a rice-based agrarian economy throughout Cambodia.





**Figure 1. Map of the Mekong River basin showing tributary and mainstream dams, hydrological stations, tributaries, and canal networks. (a) Tributaries located downstream of the Kratie gauging station that contribute to both the Mekong mainstream and Tonle Sap Lake were incorporated into the hydrodynamic model. (b) Canal systems between Kratie and the Tonle Sap confluence were included to represent flow connectivity via the Tonle Sap River in the model. Major mainstream stations and the extent of Tonle Sap Lake are also indicated.**



### 2.2 Station selection

To capture spatially representative changes in flow regime across the Mekong, this study analyzes data from eight mainstream hydrological stations, ranging from Chiang Saen in northern Thailand (representing upper-basin inflow conditions) to Kratie in Cambodia (located upstream of the Tonle Sap confluence) (Figure 1).

Additionally, we incorporated data from two supplementary stations for hydrodynamic validation and discharge lag-adjustment: Prek Kdam, situated on the Tonle Sap River just upstream of the confluence, and Kompong Luong, a monitoring station within the Tonle Sap Lake (Figure 1, part (a)). These stations were used to validate the Delft3D-Flow hydrodynamic model, which was calibrated to simulate flood inundation and reverse flow in the floodplain. (see detailed information in Sect. 2.5.1).

### 2.3 Data sources and preprocessing

Daily discharge data spanning 1976–2024 were obtained from the MRC Data Portal (https://portal.mrcmekong.org/home). Quality checks were applied to remove duplicates, missing dates, and outliers.

To assess dam-induced changes in flow, simulated daily discharge without dam influence for the post-dam period (2010–2024) was obtained from validated hydrological model simulations (see Sect. 2.5.2). Except for this case (see Figure 2d–f), it is important to emphasize that our analyses reflect the compounded impacts of dam regulation and climate variability, without explicitly disentangling their relative contributions. Accordingly, measured post-dam flows inherently capture both influences, and our results should be interpreted within this combined context.

Water level datasets – including both daily and sub-daily (15-minute interval) records – were also accessed through the MRC portal. Sub-daily water level data were available from 2018 onwards, and corresponding analyses were thus limited to this sub-period. For hydrodynamic analyses of the Tonle Sap system, additional time series of lake water levels, reverse flow periods, and discharge were acquired from the MRC and the PMFM online platform (https://pmfm.mrcmekong.org/monitoring/6b/).

To support hydrodynamic simulation of the lower Mekong–Tonle Sap–Delta system, we utilized riverbed cross-sectional data for two years: 1999 and 2018, comprising 250 measured cross sections. These datasets were obtained from MRC and other hydrological agencies. These datasets were used as bathymetric inputs for the Delft3D-Flow model (see Sect. 2.5.1). For the pre-dam and transition periods (1976–2009), the 1999 cross sections were applied, reflecting minimal sand mining-induced incision during this time. For the post-dam period (2010–2024), the more incised 2018 dataset was used to represent altered channel morphology due to intensified sand extraction. Additionally, FABDEM elevation data (Forest and Buildings removed Copernicus DEM) with 30-meter spatial resolution were incorporated to define floodplain topography and overbank areas beyond the main river channels.



## 2.4 Period classification

To capture the evolving anthropogenic impacts on the Mekong River system, we classified the historical record into three distinct hydrological periods: pre-dam (1976–1991), transition (1992–2009), and post-dam (2010–2024). The pre-dam period represents the baseline hydrological state, dominated by natural monsoonal variability and minimal human interference, with no major mainstream dams constructed during this time (Chua et al., 2022). The transition period marks the onset of large-scale regulation, with the construction of the Manwan (1992), Dachaoshan (2003), and Jinghong (2009) dams along the Lancang mainstream, alongside expansion of tributary dam networks across the lower Mekong basin (Morovati et al., 2024a). This phase also saw the emergence of relatively large-scale sand mining in Cambodia, with estimated extraction volumes reaching ~13.5 Mt yr$^{-1}$ (United Nations, 2017; Chua et al., 2022). The post-dam period is characterized by intensified regulation, marked by the commissioning of two mega-dams–Xiaowan (2010) and Nuozhadu (2014), which together increased upstream storage capacity by over 36,000 MCM. This period also featured widespread dam construction in tributaries and the expansion of irrigation infrastructure (Morovati et al., 2024a). Concurrently, sand mining intensified sharply, with annual extraction volumes in Cambodia surpassing 59 Mt yr$^{-1}$, significantly reshaping riverbed morphology and hydraulics near the Mekong–Tonle Sap confluence (Hackney et al., 2021, 2025). These compounding hydrological and morphological alterations are likely to affect not only flow magnitude and timing but also threshold-dependent dynamics such as the initiation and cessation of reverse flow into Tonle Sap Lake.

## 2.5 Methods

### 2.5.1 Hydrodynamic model

To simulate flow propagation and water surface dynamics across the lower Mekong River–Tonle Sap–Delta system, we developed a three-dimensional, grid-based hydrodynamic model using Delft3D-Flow (Deltares, 2014). The model domain spans from the Kratie gauging station in Cambodia to approximately 80 km offshore of the Sea, encompassing the Tonle Sap Lake and River, the Cambodian floodplain, and the Vietnamese delta (Figure 1).

Given the meandering nature of the Mekong River and its distributaries, we adopted a spherical coordinate system to minimize discretization errors associated with angular distortion and to ensure the accurate representation of curved channel boundaries. The model employed the cyclic method for advection and the k–ε turbulence closure scheme, which is well-suited for resolving complex three-dimensional flow structures in fluvial–estuarine systems (Morovati et al., 2023; Wu et al., 2024).

The Kratie station served as the primary upstream inflow boundary, where we imposed time-series discharge based on observed daily data. In addition, discharge time series for all major tributaries entering the Mekong mainstream or the Tonle Sap system were also defined as inlet boundaries. These were derived from a calibrated and validated physically based hydrological model (see Sect. 2.5.2). Inflow boundaries were assigned to a vertically uniform profile. The downstream open boundary was defined approximately 80 km offshore, where tidal forcing was applied using nine dominant tidal constituents



(M2, S2, K1, O1, N2, P2, K2, O2, and Q4). This tidal boundary ensured a realistic representation of tidal propagation and backwater effects from the sea into the Mekong estuarine system. The model was run with a 1-minute time step.

The horizontal grid resolution varied from 20 m in river channels and Tonle Sap tributaries to 250 m in floodplains and offshore areas. A finer resolution was employed within the river network and lake system to accurately capture channel morphology and water level gradients, both of which are critical for simulating reverse flow phenomena and calculating travel times. A one-year warm-up period was considered for simulations.

Around the Tonle Sap–Mekong confluence, many natural and artificial canals interact with the Mekong and Tonle Sap Rivers, influencing overland flow exchange during both reverse and non-reverse flow periods. These lateral exchanges, which can exceed 5 km³ yr⁻¹ (Kummu et al., 2014; Morovati et al., 2023), were fully incorporated into the hydrodynamic model. To represent these flows, we extracted canal networks using a machine learning–based remote sensing model developed by Zhao et al. (2025), supplemented by manual digitization from high-resolution satellite imagery (see Figure 1c). This enabled a more realistic representation of hydraulic connectivity between rivers, floodplains, and the lake. More details can be found in Zhao et al., 2025.

Digital elevation model (DEM) data from 250 cross-sections were integrated into the Delft3D-Flow model to define the initial bathymetric conditions. A triangular interpolation method, embedded within the model framework, was applied to river and lake cells where the DEM resolution was equal to or coarser than the model grid. For cells where the DEM resolution exceeded that of the model grid, an averaging interpolation technique was employed to preserve representative elevation values. In cells lacking direct depth data, internal diffusion interpolation was used to estimate bathymetry and ensure that every grid cell was assigned a realistic depth. This multi-step interpolation approach was designed to generate a continuous bathymetric surface that closely approximates the natural morphology of the river–lake system (see Sect. 1 in the Supplement for more details).

### 2.5.2 Response time model

To quantify the time lag between discharge at Kratie and its impact at the Mekong–Tonle Sap confluence, we developed a response time model embedded within the hydrodynamic framework described in Sect. 2.5.1.

The concept of response time, also referred to as water age, reflects the transit time required for water parcels to travel from a specified upstream point to a downstream target location (Morovati et al., 2024a). It characterizes the average residence time of water traveling from the Kratie station (inlet boundary) to the Tonle Sap confluence. This metric inherently accounts for spatial heterogeneity in flow velocity and channel morphology and thus provides a physically meaningful measure of hydrodynamic connectivity (Wu et al., 2024).

We implemented a 3D response time (age) model using an Eulerian framework, following the methodology of Shi et al. (2023). Within this framework, a virtual passive tracer–assumed to be neutrally buoyant and non-reactive–was continuously injected at the inlet boundary (Kratie). This tracer does not alter water density or momentum fields and is solely used to track





water mass movements (Morovati et al., 2024a). The response time (a) at a given location is defined as the average elapsed

time since the tracer particles first exited the inlet, weighted by their mass.

The evolution of tracer concentration (C) and age concentration (α) is governed by coupled advection–diffusion equations

derived from the principle of mass conservation:

$$\frac{\partial C}{\partial t} + (\nabla \cdot \mathrm{u})C + \frac{\partial wC}{\partial z} = \nabla(D_h \cdot \nabla)C + \frac{\partial}{\partial z}\left(D_v \frac{\partial C}{\partial z}\right) \tag{1}$$

$$\frac{\partial \alpha}{\partial t} + (\nabla \cdot \mathrm{u})C + \frac{\partial w\alpha}{\partial z} = \nabla(D_h \cdot \nabla)\alpha + \frac{\partial}{\partial z}\left(D_v \frac{\partial \alpha}{\partial z}\right) + C \tag{2}$$

$$a = \frac{\alpha}{C} \tag{3}$$

Here, C is the tracer concentration, α is the age concentration (i.e., the product of tracer concentration and residence time), D

is the turbulent diffusivity tensor, u is the velocity vector obtained from the hydrodynamic model, and a is the final computed

response time at each model grid cell. ∇ is the Hamiltonian operator.

**2.5.3 Hydrological model**

Due to the absence of measured daily discharge data for tributaries discharging into the Mekong mainstream and Tonle Sap

Lake (see Figure 1b), we employed the Tsinghua Representative Elementary Watershed (THREW) model to simulate

tributary inflows. In parallel, to generate a baseline scenario for the post-dam period (2010–2024) that is unimpacted by dam

operations, we also applied the THREW model to estimate naturalized discharge along the Mekong mainstream. The

THREW model is a spatially distributed, physically based hydrological model designed to operate at the scale of

Representative Elementary Watersheds (REWs), which preserve physical realism while ensuring computational efficiency.

The model has been extensively validated in large transboundary basins, including the Mekong (Morovati et al., 2021, 2023,

2024a; Zhang et al., 2023) (see Sect. 2 in the Supplement for more information).

The entire Mekong River basin was discretized into 651 REWs, allowing spatial representation of diverse hydrological

processes across climatic and topographic gradients (Figure S2). Site-based meteorological inputs (e.g., precipitation,

temperature) were spatially allocated to REWs using the Thiessen polygon method (Figure S3). For gridded datasets such as

Leaf Area Index (LAI) and Normalized Difference Vegetation Index (NDVI), spatial intersection analyses were performed

to determine the weighted contributions of raster cells falling within each REW. The weighted averages of these variables

were then assigned as REW-scale inputs.

Model calibration was conducted using an automatic parallel computation framework that optimizes hydrological parameters

across multiple REWs simultaneously (Nan et al., 2021) (Table S1). This enabled efficient exploration of the parameter

space while maintaining consistency with observed discharge data at available gauging stations. The calibrated THREW

model was used to simulate daily discharge for all tributaries flowing into the mainstream Mekong and the Tonle Sap Lake,

providing critical boundary conditions for the hydrodynamic simulations described in Sect. 2.5.1.





### 2.5.4 Hurst exponent: Long-term memory of flow

To quantify the persistence or memory of discharge fluctuations over time, we computed the Hurst exponent (H) using the Rescaled Range (R/S) method, a widely used method in hydrological studies (Khan et al., 2025). The Hurst exponent captures the degree of autocorrelation and long-range dependence in time series, with H > 0.5 indicating persistence, $H < 0.5$ signifying anti-persistence, and H~0.5 denoting stochastic behavior. The discharge time series $Q_t$ is divided into multiple non-overlapping segments of equal length $n$, spanning a wide range of scales. For each segment, the series is mean-centered

and transformed into a cumulative deviation series (Eq. 4), where $\bar{Q}$ is the mean discharge of the segment.

$$Y(t) = \sum_{i=1}^{t}(Q_i - \bar{Q}) \tag{4}$$

The rescaled range $R/S$ is then calculated for each segment (Eq. 5):

$$R(n) = \max(Y(t)) - \min(Y(t)), \quad S(n) = \sqrt{\frac{1}{n}\sum_{i=1}^{n}(Q_i - \bar{Q})} \tag{5}$$

Then, the average R/S is plotted as a function of window size n in log-log space (Eq. 6)

$$\log(R/S) = H \times \log(n) + C \tag{6}$$

The slope of the best-fit line yields the Hurst exponent $H \in (0,1)$. This method was implemented using the Hurst Python Package.

### 2.5.5 Richards–Baker Flashiness Index (RBI): High-Frequency Variability

The Richards–Baker Flashiness Index (RBI) was used to evaluate the short-term variability and flow "flashiness" of the

discharge series and sub-daily water levels. It is a measure of how quickly and how streamflow rises and falls in response to watershed input (Gannon et al., 2022). RBI is calculated as:

$$RBI = \frac{\sum_{i=2}^{n}|Q_i - Q_{i-1}|}{\sum_{i=1}^{n}Q_i} \tag{7}$$

where $Q_i$ is the daily discharge on day $i$, and $n$ is the number of valid observations. RBI values range from 0 (completely stable flow) to >1 (highly variable, flashy flows). This metric captures the magnitude and frequency of day-to-day discharge

changes, allowing assessment of how dam regulation has attenuated or enhanced hydrograph variability.

### 2.5.6 Coefficient of Variation (CV): Interannual discharge variability

To assess changes in relative flow variability across the basin, we computed $CV$ for each station and time period as (Blöschl and Sivapalan, 1997; Li et al., 2025):

$$CV = \frac{\sigma Q}{\mu Q} \tag{8}$$

where $\sigma_Q$ and $\mu_Q$ denote the standard deviation and mean of daily discharge, respectively. This dimensionless index provides a standardized measure of variability that facilitates comparisons between stations with differing flow magnitudes.





## 3 Results

### 3.1 Model validation

The performance of the THREW and Delft3D-Flow models was evaluated across multiple hydrological stations. Both models demonstrated high accuracy in reproducing discharge and water level time series, with Nash–Sutcliffe Efficiency (NSE) values exceeding 0.88. Detailed validation results and station-specific performance metrics are provided in the Supplement, Sects. 2 and 3.

### 3.2. Multi-scale streamflow alterations under dam regulation and climate change

Six rose diagrams (a-f) summarize the spatiotemporal evolution of discharge dynamics across eight mainstream stations
from Chiang Saen to Kratie. Top-row panels show observed changes across three hydrological periods, while bottom-row panels compare measured vs. no-dam scenario for the post-dam era (2010–2024). All metrics are derived from the daily discharge series.

The Hurst exponent (H) captures long-term persistence in discharge variability (Figure 2a). Values exceed 0.9 at Chiang Saen and steadily decline downstream, stabilizing near 0.79 at Kratie. This upstream-to-downstream gradient is consistent
across all three periods, with the steepest decline observed during the post-dam era. The marked increase in H at upstream stations (e.g., Chiang Saen: 0.80 to 0.93) suggests that cascade dams, particularly Lancang dams and downstream hydropower infrastructure, have enhanced streamflow persistence by dampening short-term fluctuations. In contrast, downstream stations exhibit limited changes in Hurst values between pre- and post-dam periods, likely due to the compensating effects of tributary inflows. The calculated H values for the post-dam period by measured data are consistently
higher than the no-dam scenario (e.g., +0.13 at Chiang Saen), reinforcing the conclusion that dam operations amplify flow memory by suppressing stochastic discharge variability (Figure 2d).

The RBI quantifies day-to-day discharge fluctuations (Figure 2b). Although the overall spatial pattern of RBI remains relatively consistent between the pre-dam and transition periods, noticeable deviations are observed at certain stations, such as Chiang Saen and Kratie, suggesting emerging regulatory influences or climatic variability even before the onset of large-
scale dam operations in the post-2010 era. However, the post-dam period introduces a spatially bifurcated pattern: RBI declines markedly in the upper basin (e.g., Chiang Saen drops from ~0.36 to ~0.25), reflecting pronounced flow smoothing from upstream reservoir regulation, while downstream stations retain relatively higher values due to the influence of tributary inflows and attenuated dam effects. This pattern reflects the dual influence of dam-induced flow smoothing upstream and tributary-driven variability downstream. The measured vs. no-dam comparison confirms this pattern (Figure
2e). RBI values are lower under the measured scenario at all stations, with upstream reductions of 25–30% (Figure 2e).

The CV captures the amplitude of seasonal and interannual discharge fluctuations. CV values remain high during the pre-dam and transition periods, particularly downstream of Nakhon Phanom, but decline substantially in the post-dam period at upstream sites (e.g., Chiang Saen drops from 0.68 to ~ 0.52), consistent with zones of maximum reservoir influence (Figure



2c). Comparison with no-dam simulations reinforces this pattern (Figure 2f): measured CV values are systematically lower
than their no-dam counterparts at upstream stations (e.g., Chiang Saen: 0.52 vs. 0.68; Chiang Khan: 0.71 vs. 0.81),
confirming that reservoir operations have dampened both seasonal amplitude and interannual variability. In contrast, at
downstream stations such as Mukdahan, Pakse, Stung Treng, and Kratie, CV differences between measured and no-dam
scenarios are minimal, suggesting that the buffering effects of upstream regulation diminish with distance or are offset by
inflows from unregulated tributaries and floodplain dynamics.


**Figure 2. Spatial patterns of streamflow memory, flashiness, and variability across the Mekong River under dam-
regulated and no-dam scenarios. Six rose diagrams summarize hydrological dynamics at eight mainstream stations
from Chiang Saen to Kratie. The top row (a–c) compares three hydrological periods–pre-dam (1976–1991), transition
(1992–2009), and post-dam (2010–2024)–for (a) the Hurst exponent (H), (b) Richards–Baker Index (RBI), and (c)
Coefficient of Variation (CV). The bottom row (d–f) compares post-dam conditions under measured vs. no-dam
simulations. Note that the no-dam scenario simulations were derived using the THREW hydrological model. The
term "measured" in panels (d–f) refers to observed mainstream flows shaped by the combined influence of upstream
dam regulation and climate change.**





### 3.3 Spatial patterns of sub-daily flow variability

Figure 3 illustrates sub-daily water level dynamics across the lower Mekong using three complementary metrics: amplitude, flashiness (RBI), and peak count ($\geq 5$ cm hr$^{-1}$). The peak count metric quantifies the number of hourly water level rises that exceed the 5 cm hr$^{-1}$ threshold, which the MRC (MRC, 2020) identifies as the maximum allowable rate of change to minimize downstream ecological impacts of dam operations. Stations such as Chiang Khan and Pakse exhibit pronounced sub-daily amplitudes during the wet season, with values exceeding 0.6 m in June–July at Chiang Khan and reaching a peak

of 0.785 m at Pakse in September. In contrast, downstream stations display more subdued fluctuations: Stung Treng maintains amplitudes below 0.27 m year-round, while Kratie – functioning as a critical hydrological node for the Tonle Sap system – registers a notable increase to 0.46 m in September, coinciding with the peak reverse flow period (Figure 3a).

Monthly flashiness values at Pakse peaked at 0.011 in September – the highest recorded among all sites – indicating intense and frequent sub-daily variability (Figure 3b). Nakhon Phanom exhibited moderately elevated RBI values during the wet

season (~0.001–0.005), whereas stations such as Chiang Khan and Nong Khai remained below 0.003 throughout the year. In contrast, downstream stations, including Stung Treng and Kratie, consistently recorded RBI values below 0.001, indicative of attenuated flashiness and smoother diurnal dynamics. The spatial coherence between amplitude and RBI patterns delineates a clear upstream-to-downstream gradient in flow responsiveness, likely governed by the interplay of regulated hydropower releases and monsoon-driven inflows.

Pakse stands out with a monthly average of 9.37 peaks/day in September. Upstream stations such as Chiang Khan and Nakhon Phanom also show elevated peak counts exceeding 1 event/day during wet months, while most other stations and months exhibit low or negligible values, suggesting that such fluctuations are episodic and spatially localized (Figure 3c).

Annual trends across all three metrics corroborate the spatial gradient in sub-daily variability. Stations such as Chiang Khan, Nakhon Phanom, and Pakse consistently exhibit heightened sub-daily variability, while downstream locations, as well as

Chiang Saen, the most upstream station situated immediately downstream of the Lancang cascade, exhibit comparatively muted signals, highlighting the attenuation of intraday fluctuations along the river continuum.





**Figure 3. Sub-daily water level variability across eight Mekong mainstream stations (2017–2024), assessed using three metrics: amplitude (a), flashiness index (RBI) (b), and peak count ≥ 5 cm hr$^{-1}$ (c). Note: amplitude, defined as the daily range between maximum and minimum water levels, exhibits strong spatial contrasts. Sub-daily water level data were only available from 2018-2024.**



## 3.4 Changes in annual discharge extremes and their timing

A pronounced reduction in annual peak discharge is evident at several stations following upstream dam development (Figure 4a). At Chiang Saen, which reflects transboundary inflow from the Lancang cascade, the median peak flow declined by ~ 40%, from ~10,850 m³ s⁻¹ in the pre-dam period to ~6,493 m³/s in the post-dam era. Similar, though less pronounced, reductions in median peak discharge are observed at Chiang Khan and Nong Khai, where values declined by approximately 11.4% and 8.4%, respectively, from the pre-dam to post-dam period. These trends reflect attenuated flood pulses downstream of the Chinese cascade and possibly early regulatory effects from tributary inflows. In contrast, Nakhon Phanom exhibited a 9.6% increase in peak discharge, suggesting localized hydrological intensification, potentially influenced by regional rainfall variability or unregulated tributary contributions. At Kratie–a hydrologically strategic station controlling the Mekong inflow to the Tonle Sap River and lake during the wet season months–the median annual peak discharge declined from ~ 38,300 m³ s⁻¹ in the pre-dam period to 34,700 m³ s⁻¹ in the post-dam period, representing a 9.4% reduction.

Conversely, annual minimum discharges exhibit a consistent increasing trend across all stations, particularly during the post-dam period (Figure 4c). At Chiang Saen, the median minimum discharge rose from approximately 748 m³ s⁻¹ in the pre-dam period to 997 m³ s⁻¹ in the post-dam period–an increase of 33.3%, reflecting enhanced baseflow likely sustained by regulated dam releases. This upward trend is even more pronounced downstream. At Stung Treng, minimum flow increased from 1,565 m³ s⁻¹ to 2,820 m³ s⁻¹, a remarkable 80.2% rise. Most notably, at Kratie, minimum discharge surged from 1,326 m³ s⁻¹ in the pre-dam period to 2,880 m³ s⁻¹ in the post-dam period, marking an increase of 117.2%. These substantial increases in dry-season flow highlight the transformative impact of upstream tributary and mainstream reservoir storages and regulated releases on the seasonal amplitude of Mekong River discharge, potentially compressing the natural flood pulse regime and altering the hydro-ecological dynamics of the floodplain and Tonle Sap system.

In the pre-dam period, peak flows generally occurred between early August and early September across all stations, aligned with the core of the monsoon season (Figure 4b). However, dam regulation and reservoir operations have systematically delayed the timing of peak flows in the post-dam period. Peak flow timing at most stations has shifted by approximately 10–20 days later. At Kratie, the median day of peak flow occurred around late August, whereas in the post-dam period, this timing has shifted to approximately mid-September.

The timing of annual minimum flows has consistently advanced across all stations along the Mekong mainstream. At Kratie, for example, the timing of median minimum discharge shifted from early April (April 10) to mid-March in the post-dam period, an advancement of nearly a month. Similar shifts are observed throughout the basin, with Chiang Saen advancing from March 25 to March 2, Chiang Khan from April 9 to March 14, Stung Treng from April 9 to March 11, and from April 10 to March 12. These 2–4-week advancements in low-flow timing suggest a system-wide shift in hydrological seasonality, likely driven by altered monsoon onset patterns and the compounded influence of reservoir discharge strategies in both the upper and lower Mekong basin.





**Figure 4. Alterations in annual discharge extremes and their timing across the Mekong mainstream. Boxplots show the spatial and temporal evolution of (a) annual maximum daily discharge, (b) timing of annual maximum discharge, (c) annual minimum daily discharge, and (d) timing of annual minimum discharge.**

## 4 Discussion

While moderate alterations in the flow regimes of large rivers and their floodplains are anticipated, particularly in tropical monsoonal regions undergoing rapid agricultural expansion and extensive dam construction – disrupted flow memory and fragmented synchrony represent deeper hydrological disturbances that threaten the ecological integrity and aquatic productivity of river–lake systems. Despite numerous prior studies on the Mekong, this work offers new insights by uncovering a breakdown in both temporal hydrological persistence and spatial flow coherence, highlighting a fundamental shift in the basin's natural dynamics. The suppression of short-term variability and flattening of intra-annual discharge pulses–evidenced by rising Hurst exponents and declining RBI values, particularly in the upstream basin–disrupt the natural flow cues that many aquatic species rely upon. In the Mekong, flood-dependent fish species synchronize their spawning,



migration, and feeding behaviors with the onset and magnitude of seasonal flow pulses. Attenuation of these pulses weakens floodplain–mainstream connectivity, restricts access to critical spawning and nursery habitats, and reduces the heterogeneity of aquatic environments that underpin the basin's exceptional biodiversity (Baran and Myschowoda, 2009; Arias et al., 2014;

MRC, 2017; Dudgeon, 2020). Thus, the observed fragmentation in hydrological synchrony and decline in discharge flashiness may not only diminish the ecological productivity of the river–lake system but also erode its resilience to future climatic and anthropogenic stressors.

The findings on flow regime changes at mainstream stations are consistent with previous studies, despite differences in the temporal resolution of analysis (e.g., monthly vs. annual). A similar pattern of discharge alteration is observed (Li et al.,

2017; Räsänen et al., 2017; Yun et al., 2020). For instance, Li and Chua (2021) reported a 33.7% increase in the annual minimum discharge at Chiang Saen during 2010–2020 relative to the 1960–1991 period. This study finds a nearly identical increase (33.3%) for the extended post-dam period of 2010–2024. This suggests that even with the inclusion of the most recent four years, upstream regulatory impacts have not further altered minimum flows at Chiang Saen, indicating a potential stabilization of baseflow conditions under current dam operation regimes.

**4.1 Contraction of the Tonle Sap reverse flow period**

This study also bridges basin-scale discharge changes with the reverse flow mechanism at the Mekong–Tonle Sap confluence, offering new insight into the timing, magnitude, and propagation dynamics that govern seasonal water redistribution. Figure 6a highlights two critical elements: (i) the nonlinear discharge–travel time relationship between Kratie and the confluence, and (ii) the evolving monthly discharge regimes under different dam-development periods. During the

wet season (June–November), high flows at Kratie typically reach the Tonle Sap confluence within 1–3 days, as indicated by the fitted power-law relationship $RT = 1060.6 \times Q^{-0.629}$. This short response time has important operational implications, particularly for early warning systems. For example, during flood-prone years such as 2011, the rapid propagation of floods from Kratie to Phnom Penh and the delta increased disaster vulnerability (Chinh et al., 2016). With the ongoing intensification of hydropeaking and discharge variability (Figures 2-4), accurate estimates of travel time become increasingly

relevant for downstream risk preparedness.

Concurrently, the inset bar plot (Figure 5a) reveals a dampened and more peaked monthly hydrograph in the post-dam period. While September remains the peak month across all periods, the early wet season flows in June and July have diminished, and the rise toward the peak has become more abrupt. This altered hydrograph shape contributes to a temporal decoupling between early-season Mekong inflow and reverse flow onset, weakening the natural synchrony that historically

governed the Tonle Sap Lake's seasonal flooding regime.

Panel 5b further dissects this relationship by showing lag-adjusted discharge thresholds at Kratie associated with the onset and cessation of reverse flow into the lake. In the pre-dam period (1976–1991), reverse flow typically began when discharge exceeded ~3,000 m³ s⁻¹ and ended around ~28,000 m³ s⁻¹. In contrast, during the post-dam period (2010–2024), the onset threshold increased by 130% to ~7,000 m³ s⁻¹, while cessation now requires discharges exceeding ~34,000 m³ s⁻¹, with some



years surpassing 40,000 m³ s⁻¹. These shifts reflect a significant hydraulic reconfiguration of the system. Two primary drivers are implicated:

   (i) Riverbed degradation and sand mining at the confluence have deepened the channel and increased the hydraulic head required to reverse Tonle Sap River flow (Hackney et al., 2025).

   (ii) Greater short-term discharge variability at Kratie, as shown in Figures 2-4, introduces instability in the gradient
430       dynamics needed to sustain reverse flow, requiring more persistent and elevated upstream flows to compensate.

Panel 5c underscores the temporal reconfiguration of reverse flow dynamics. Historically, the onset of reverse flow occurred in early June, with a median start date around June 3–5 during the pre-dam period. In the post-dam period, this onset has been delayed by approximately two weeks, now occurring around June 17 on average. Conversely, the cessation of reverse flow, which historically occurred in the first ten days of October, has now advanced by nearly two weeks, with most years
showing end dates around the third decade of September. These shifts point to a shortened reverse flow season, primarily driven by delayed upstream flood arrivals and earlier downstream drawdown conditions. This altered timing aligns with broader basin-scale changes observed in Figures 2 and 4, where the peak discharge has shifted to September and flood durations have contracted, particularly under intensified post-dam regulation.

Altogether, Figure 5 supports the growing body of evidence that hydrological connectivity between the Mekong and Tonle
Sap Lake is weakening, shaped by both anthropogenic (dam operation, sand extraction) and climatic factors (delayed monsoon onset, flashier flows). These shifts reduce the volume and duration of lake inflow, threatening floodplain productivity and the resilience of the Tonle Sap socio-ecological system. Importantly, this study introduces a diagnostic approach linking lag-adjusted thresholds to reverse flow timing, which can be integrated into basin-wide early warning systems and adaptive flow regulation frameworks under future climate and dam development scenarios.



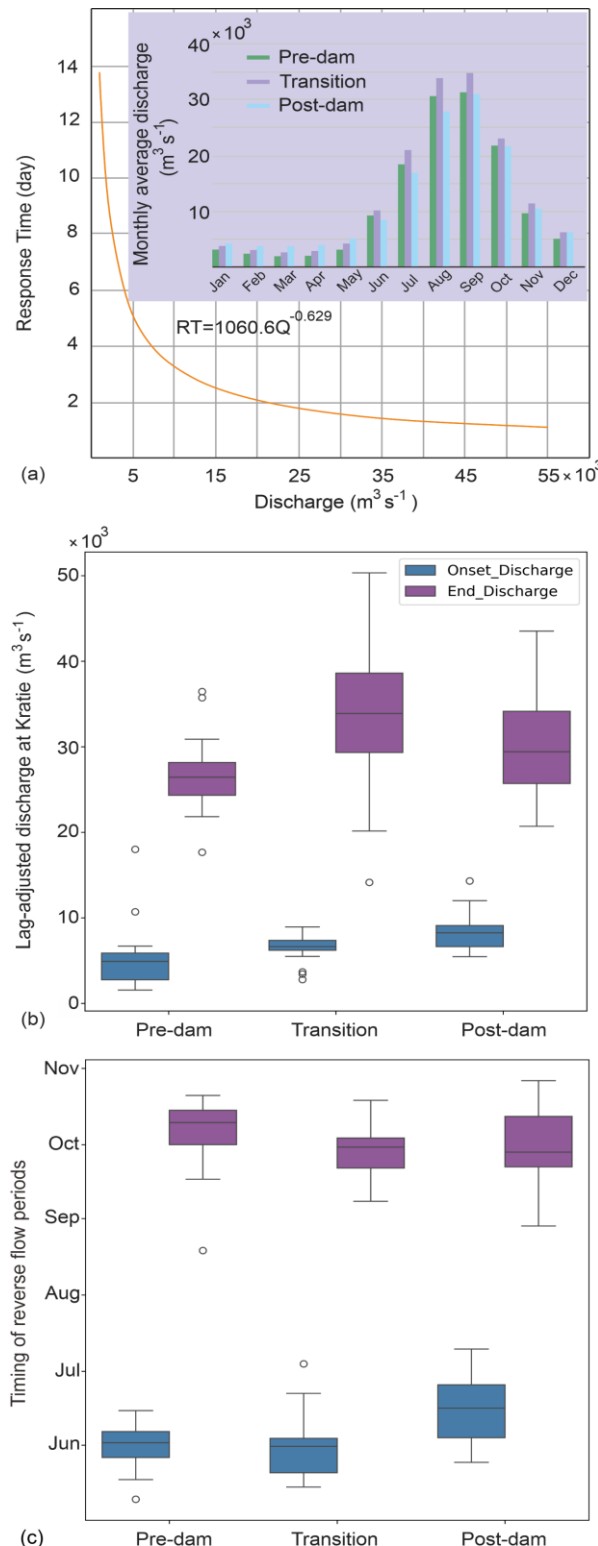






**Figure 5. (a) Estimated travel time (days) for Kratie discharge to reach the Tonle Sap–Mekong confluence, overlaid with monthly average discharge at Kratie. (b) Lag-adjusted discharge thresholds at Kratie corresponding to the onset (left) and cessation (right) of reverse flow into Tonle Sap Lake. (c) Timing of reverse flow onset (left) and cessation (right). Note that historical data on the Tonle Sap Lake's reverse flow period are available from 1997 onward**
**([https://pmfm.mrcmekong.org/monitoring/6b/](https://pmfm.mrcmekong.org/monitoring/6b/), last access: 16 July 2025). Consequently, we reconstructed reverse flow periods from 1976 to 1996 using a calibrated hydrodynamic model (see Sect. 3 in Supplements)**

### 4.2 Limitations and ways forward

While this study provides a holistic analysis of hydrological alterations across both long-term and sub-daily scales, several limitations remain that warrant further investigation. First, the attribution of observed sub-daily fluctuations to upstream
hydropower operations is inherently constrained by the absence of publicly available dam release data. Although the temporal alignment between wet-season peaks and sub-daily variability supports a linkage to hydropeaking, direct validation through operational records remains essential.

The interaction between river morphology, sediment loads, and hydraulic thresholds for reverse flow remains underrepresented. While the observed elevation in Kratie discharge thresholds aligns with regional bed incision and sand
mining, spatially explicit bathymetric and sediment transport data are needed to robustly quantify their role.

While the comparison between measured (with-dam) and simulated no-dam scenarios reveals important shifts in flow memory, flashiness, and synchrony, it is important to acknowledge that uncertainty exists in the hydrological model used to simulate the no-dam scenario. Model uncertainties may influence flow amplitude and timing. Future work should quantify these uncertainties through ensemble simulations or data assimilation techniques.
This study focuses primarily on hydrological metrics and does not explicitly assess ecological or livelihood outcomes. Integrating these flow regime alterations with floodplain vegetation dynamics, fish migration cues, and agricultural timing would provide a more comprehensive view of cascading socio-ecological impacts.

### 5 Conclusion

This study provides a comprehensive spatiotemporal assessment of hydrological alterations in the Mekong River system,
identifying extensive dam construction on both the mainstream and tributaries, along with climate change, as the primary drivers of observed changes. In the Tonle Sap system, sand mining emerges as an additional, significant anthropogenic stressor.

Hydrological metrics – including the Hurst exponent, flashiness index, and coefficient of variation – reveal that the Mekong mainstream is no longer functioning as a hydrologically coherent system. Regulation, dam sequencing, and other
interventions have fragmented its natural connectivity, particularly during the post-2010 mega-dam era. The synchrony of





seasonal flow patterns has weakened along the river's longitudinal profile, with fragmented flow dynamics replacing the previously unified flood pulse.

At Chiang Saen, the closest station to the Lancang cascade dams, annual minimum flows increased by 33.3% (from 748 m³ s⁻¹ to 997 m³ s⁻¹) in the post-dam period. A similar trend is observed downstream, most notably at Kratie, where minimum flows rose by 117.2% (from 1,326 m³ s⁻¹ to 2,880 m³ s⁻¹). In contrast, annual maximum flows decreased by 40% at Chiang Saen and by only 9.4% at Kratie, likely due to the buffering effects of monsoonal tributary inflows and dam operations on these tributaries.

These mainstream alterations have propagated into the Tonle Sap system. The reverse flow period has shortened by approximately 24 days during the post-dam period compared to the pre-dam baseline. Additionally, reduced Mekong high flows, compounded by riverbed incision from sand mining, have significantly raised the discharge threshold required to initiate reverse flow – from a median of ~3,000 m³ s⁻¹ (pre-dam) to ~7,000 m³ s⁻¹ (post-dam), representing a >130% increase. The combined evidence of dampened high flows, intensified sub-daily variability, and shifting seasonal flow timings points to a river system undergoing profound anthropogenic transformation. As regional development intensifies, future management must extend beyond annual water volume considerations, focusing instead on the temporal structure and synchrony of flow regimes – key to sustaining downstream ecosystems, floodplain agriculture, and rural livelihoods. Preserving the integrity of the Mekong–Tonle Sap flood pulse will require basin-wide hydrological monitoring, transparent data sharing, and coordinated transboundary water governance.

**Code and data availability.** Hydrological data, including water level and discharge data, are accessible at https://portal.mrcmekong.org/home (last access: 16 July 2025). Code for extracting canals across the lower Mekong can be obtained upon reasonable request from the primary author of the paper. FABDEM data is publicly available at https://www.fathom.global/product/global-terrain-data-fabdem/ (last access: 16 July 2025). Constructed DEM data for Tonle Sap Lake and rivers can be obtained upon reasonable request from the primary author of the paper. Data of the reverse flow period is publicly available at https://pmfm.mrcmekong.org/monitoring/6b/ (last access: 17 July 2025).

**Author contributions:** KM conceptualized and designed the study. KM, HZ, and FT contributed to model development, with KM and HZ responsible for model implementation and analysis. KM led the manuscript writing in close collaboration with HZ and FT. All authors contributed to discussions throughout the study, provided critical feedback, and approved the final version of the manuscript.

**Competing interests.** At least one of the (co-)authors is a member of the editorial board of Hydrology and Earth System Sciences.

**Acknowledgment.** This research was supported by the Shuimu Tsinghua Scholar Program. The authors confirm that no portion of this manuscript was generated or assisted by AI tools.



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
