# Peer review of "Disrupted Flow Memory and Synchrony in the Mekong River under Dam Regulation and Climate Change: Implications for Tonle Sap Reverse Flow"

_EGUsphere, 2025_

## Author Comment (AC1)

We thank the anonymous reviewer for the constructive suggestions and comments. Below, we provide point-by-point responses to each comment. Our replies are introduced by "Response:". Text highlighted in blue indicates revisions that are incorporated into the revised manuscript.
* * *
**The study provides a comprehensive and timely analysis of the alterations in flow regime in the Mekong mainstream and their intricate, yet crucial linkage with the Tonle Sap Lake system, both of which have entered a critical phase of change under dam regulation and climate change. This study offers a new perspective on the changes in river flow regime and river-lake connectivity through alternative hydrological metrics, flow extremes, and the response time, as well as reverse flow periods. While the methodology is robust and the conclusions are generally well-supported, several key concerns require clarification and improvement for better readability and scientific rigor. Please find my specific and minor comments as follows.**

**Response:** The authors thank the reviewer for the positive comments, feedback, and suggestions. Please find our replies to each comment below, which show how the authors want to consider the comments in the revised manuscript.

**Specific comments:**

**1. L109-L113: Since the study importantly points to the Mekong River-Tonle Sap Lake dynamics, more numerical details on the hydrology/hydrodynamics of the lake are necessary. For instance, Tonle Sap Lake is also contributed by the Tonle Sap tributaries to an extent that is, however, less than the reverse flow contribution.**

**Response:** We thank the reviewer for this helpful suggestion. We agree that, given the central role of Mekong–Tonle Sap coupling in our study, the manuscript should provide clearer quantitative context on Tonle Sap Lake hydrology and the relative magnitude of its inflow components. Accordingly, we add the following information to the revised manuscript.

*Downstream, the Mekong River interacts seasonally with the Tonle Sap Lake via the Tonle Sap River. During the wet season, strong mainstream flows cause a hydraulic gradient reversal, pushing water back into the lake and expanding its surface area dramatically—from approximately 2,500 km² to over 13,000 km². Lake water level typically varies from ~ 1.2 to 10.4 m (Dang et al., 2022), corresponding to storage changes of 1.6–59.7 km³ (Kummu et al., 2014). Water-balance analyses indicate that ~42-53.5% of annual inflow originates from the Mekong mainstream, whereas the lake's tributaries contribute ~34-41% and direct precipitation ~12.5% (annual inflow range 51–109 km³; mean ~83.1 km³) (Morovati et al., 2023, Kummu et al., 2014).*

**2. Section 2.2: The section did not list Phnom Penh Port station, while Figure 1 depicts it. What is the role of the station in the study? Even if the station's location is used, not the hydrological data, how it was used should be clarified.**

**Response.** Thank you for pointing this out. Phnom Penh Port was shown in Fig. 1 inadvertently; it was not used in our flow-regime analyses. To avoid confusion, we have removed Phnom Penh Port from Fig. 1 and updated the figure caption accordingly.

[Figure]

**3. Section 2.3: Weather data for the THREW model were not introduced. Please provide the details and sources of all input data for the hydrological and hydrodynamic models.**

**Response:** Thank you for this comment. We agree that the meteorological forcing and other THREW input datasets should be explicitly documented. We revise Section 2.3 to provide the variables and sources used to force THREW (precipitation, temperature, and Penman–Monteith potential evapotranspiration), as well as the land-surface/vegetation datasets used for parameterization (soil properties and MODIS-based vegetation/snow products):

In section 2.3, we add the following details:

*The THREW hydrological simulations were driven by station-based precipitation and meteorological observations from the Mekong River Commission (MRC) and the China Meteorological Administration (CMA). Precipitation was obtained from a basin-wide gauge network (105 stations) and air temperature from 35 stations (Fig. S3). Daily potential evapotranspiration was computed using the Penman–Monteith method based on station meteorological variables (including temperature, wind speed, humidity, and*

*radiation/sunshine duration; Fig. S3). Soil properties were taken from the FAO global soil database (10 km). Vegetation and surface-condition inputs (NDVI, LAI, and snow cover) were derived from MODIS products (500 m, 16-day) following Zhang et al. (2023).*

**4. Section 2.4: This section should appear before Data Sources and Preprocessing, as it gives precedent information on changing morphology in the segregated periods.**

**Response:** Thanks for your good suggestion. We agree. Because the period classification governs both the interpretation of regime shifts and our period-specific representation of channel morphology (e.g., selection of 1999 vs 2018 cross-sections), we move the period-classification section to appear before 'Data sources and preprocessing' to improve logical flow and readability. Thank you.

**5. L187: Is a one-year warm-up period good enough to initialize the model, given the complex system? Should there be any potential limitations pertaining to this setup?**

**Response:** Thank you for raising this point. We agree that the adequacy of a warm-up period should be justified for a coupled river–lake–floodplain system. We used **a one-year warm-up** because it comfortably exceeds the dominant **hydrodynamic adjustment timescales** of the system and includes a full seasonal cycle (dry-to-wet transition, flood rise, and recession), which is essential for initializing storage and exchange fluxes.
First, our response time analysis (Fig. 5a) indicates that the propagation time of mainstream flow from Kratie to the confluence is ~1–14 days, depending on discharge. This short hydraulic response implies that boundary perturbations and initial condition effects are rapidly flushed from the river network relative to a one-year spin-up.
Second, the lake–floodplain component requires initialization of seasonal storage dynamics. A one-year warm-up explicitly contains one complete flood pulse, allowing lake level, inundation extent, and exchange flows to adjust consistently to the model physics and boundary forcing. Consistent with this, our prior Tonle Sap Lake modelling work used a 3-month warm-up and found it sufficient for stabilizing lake dynamics, even though that domain was smaller than in the present study (Morovati et al., 2023). The longer one-year warm-up adopted here is therefore a conservative choice.
Third, we empirically verify adequacy by examining model performance **immediately after the warm-up year**. If the spin-up were insufficient, the following year would typically show systematic bias or transient drift in simulated water levels. However, the model achieves high accuracy in 2010 when 2009 is treated as the warm-up year (Supplementary Fig. S5), indicating that initial-condition sensitivity has largely decayed.

**6. Sections 2.5.1, 2.5.2, and 2.5.3: I believe the order of the three sections should be reorganized, as the development of the hydrological model is crucial as the boundary condition for the hydrodynamic model, and then the response model was embedded in the hydrodynamic framework. This is also in line with the order of the analyses.**

**Response:** Thank you for this helpful suggestion. We agree and reorder Sections 2.5.1–2.5.3 to follow the model dependency chain: THREW (hydrological) → Delft3D-Flow (hydrodynamic) → response-time (age) model. We also correct internal cross-references so that

tributary inflows and naturalized discharge are now consistently referenced to the hydrological model section.

**7. Section 2.5.2: The response time model is for the stretch between Kratie and the Mekong-Tonle Sap confluence, which is in Phnom Penh, specifically at the Phnom Penh Port location. However, it was stated in Section 2.2 that Prek Kdam and Kompong Loung were listed for hydrodynamic validation and also discharge lag adjustment, implying that either station was considered for adjusting the discharge lags. Please clarify both sections.**

**Response:** Thank you for noting this. The response-time (water-age) model quantifies lag from Kratie to the Mekong–Tonle Sap confluence (Chaktomuk) (line 204), which is the hydraulic control point governing whether the Tonle Sap River reverses direction. The Phnom Penh Port gauge is nearby but not identical to the junction (located on the Tonle Sap River ~4 km from the confluence).

Prek Kdam and Kompong Luong were included for hydrodynamic validation and for characterizing river–lake exchange, **not as the target location for lag adjustment.** We revise this Section to explicitly distinguish (i) the lag-adjustment target at the confluence from (ii) validation stations (Prek Kdam on the Tonle Sap River and Kompong Luong in the lake).

In the revised hydrodynamic section, we can mention that two stations are used for hydrodynamic validation (Prek Kdam on the Tonle Sap River and Kompong Luong in the lake).

**8. L288: Higher monsoonal rainfall predominantly in the lower Mekong can also be highlighted in addition to tributary inflows.**

**Response:** We agree and thank the reviewer for this helpful suggestion. In the original text (L288), we attributed the relatively muted downstream changes primarily to compensating effects of tributary inflows. We revise this sentence to explicitly note that these downstream tributary and floodplain contributions are predominantly monsoon-driven, and that intense wet-season rainfall in the lower Mekong enhances local runoff and can partially offset (or mask) the upstream regulation signal. The revised text reads as follows:

*..... likely due to the compensating effects of monsoon-driven runoff contributions from downstream tributaries and adjacent floodplains, which become increasingly important in the lower basin.*

**9. Section 3.3: Some of the methods, like amplitude and peak count, should be briefly explained prior in the Method section.**

**Response:** Thank you for this helpful suggestion. We agree that the definitions of the sub-daily water-level metrics should be provided in the Methods for clarity and reproducibility. In the revised manuscript, we added a short subsection in Section 2.5 describing the computation of (i) sub-daily amplitude (daily max–min water level range), (ii) peak count (number of hourly water-level rises $\geq 0.05$ m h$^{-1}$), and (iii) the implementation of RBI for sub-daily water level time series. This ensures that all metrics used in Section 3.3 are defined consistently before presentation of results. The revised subsection reads as follows:

*Sub-daily water-level variability metrics*

*Sub-daily water-level variability was quantified using three complementary metrics derived from 15-min water level records (available from 2018 onward): (1) amplitude, defined for each day as the difference between the daily maximum and daily minimum water level ($A\_d = WL\_max,d − WL\_min,d$); monthly values were computed as the mean of daily amplitudes within each month. (2) flashiness (RBI) computed for the sub-daily water-level time series using the Richards–Baker formulation (Eq. 7) by substituting WL for Q and using consecutive sub-daily observations within each month to obtain a dimensionless index of intraday variability. (3) peak count, defined as the number of hourly water-level rises exceeding a threshold of 0.05 m h$^{-1}$; 15-min records were aggregated to hourly water levels, and an event was counted when $\Delta WL/\Delta t \geq 0.05$ m h$^{-1}$ for a positive rise. Peak count is reported as peaks per day and averaged by month.*

**10. Figure 3: The figure and its caption mention 2018-2024, but the caption also states 2017-2024. Could you elaborate on the difference?**

**Response:** Thank you for pointing this out. The correct period is 2018–2024. We revise the figure and its caption accordingly.

**11. Section 4.1: I find that this discussion section provides new major results along with their discussion. Those results are directly relevant to the topic and objectives. It makes more sense to transition those findings to the Results section; hence, this leaves more room for their implications to be expanded in the Discussion. Equally important is that previous studies should be cited in the section to enhance the interpretation of results and discussion, as the Mekong-Tonle Sap Lake connectivity has been increasingly explored with a wide range of implications beyond hydrodynamics.**

**Response:** Thank you for this helpful suggestion. We agree that the original Section 4.1 mixed new quantitative findings with interpretation. In particular, the travel-time relationship between Kratie and the confluence and the period-wise changes in lag-adjusted discharge thresholds and reverse-flow timing were first presented in the Discussion.
We move these quantitative findings to the Results as a new subsection (**Section 3.5:** *Travel-time-adjusted discharge thresholds for reverse-flow onset and cessation*), and we revise Section 4 to focus on interpretation and implications. In the revised Discussion, we also expand citation of previous work on Mekong–Tonle Sap connectivity and its broader implications.

**Minor comments:**

**1. L24-25: It is better to mention "one of the world's most ecologically productive river–lake systems" early in the Abstract to highlight the significance of the study area.**

**Response:** Thank you for this suggestion. We agree and revise the opening sentence of the Abstract to highlight the global ecological significance of the Mekong–Tonle Sap Lake system, which reads as follows:

*Dam construction and climate change have profoundly disrupted the hydrological dynamics of the Mekong River–Tonle Sap Lake floodplain system, one of the world's most ecologically productive river–lake complexes. This study provides an integrated, .......*

**2. L45 and L47: Please clearly indicate the months in the wet and dry seasons.**

**Response:** Thank you for the suggestion. We agree that the seasonal definitions should be stated explicitly. The revised text reads as follows:

*These impacts have been especially pronounced during the dry season (November–April). Räsänen et al. (2017) reported dry-season discharge increases of 121–187% at Chiang Saen in March and 32–46% at Kratie, while Lu and Chua (2021) found a 98% increase in monthly discharge at Chiang Saen during the dry months. In contrast, wet-season flows (May–October) have declined substantially (Lu et al., 2014), weakening the magnitude and altering the timing of flood pulses that sustain floodplain ecosystems.*

**3. L46: The relative locations of Chiang Saen and Kratie stations should be addressed for the first time (i.e., the most upstream and downstream stations).**

**Response:** Thank you for the comment. We revise the text at L46 to clarify the relative locations of Chiang Saen and Kratie (upstream vs downstream endpoints of the study reach) and added an explicit reference to Fig. 1, which reads as follows:

*These impacts have been especially pronounced during the dry season (November–April). Räsänen et al. (2017) reported dry-season discharge increases of 121–187% at Chiang Saen (uppermost station) in March and 32–46% at Kratie (lowermost station) (Fig. 1), while Lu and Chua (2021) found a 98% increase in monthly discharge at Chiang Saen during the dry months. In contrast, wet-season flows (May–October) have declined substantially (Lu et al., 2014), weakening the magnitude and altering the timing of flood pulses that sustain floodplain ecosystems (Fig. 1).*

**4. L80: For the last paragraph, I noticed that sub-daily variability was not indicated, although it was analysed with its own subsection in the Results. It should be listed here in the Introduction for a full picture of the objectives.**

**Response: Thanks for this good comment. The revised text reads as follows:**

*This study addresses these gaps through a multi-decadal, multi-indicator analysis of flow regime shifts at eight mainstream stations along the Mekong River from 1976 to 2024. We quantify changes in daily discharge variability, flashiness, and memory across three hydrological periods: pre-dam (1976–1991), transition (1992–2009), and post-dam (2010–2024), introducing the concepts of disrupted flow memory and fragmented synchrony to describe the breakdown in spatiotemporal coherence. In addition to these indicators, we examine annual discharge extremes, including maximum and minimum daily flows and their associated timing. Finally, using 15-min water-level observations (2018–2024), we quantify sub-daily variability using amplitude, flashiness (RBI), and peak count metrics. We further link these altered flow characteristics to the onset, termination, and duration thresholds of Tonle Sap's reverse flow using a Delft3D-Flow hydrodynamic model to estimate Kratie-to-confluence response time and apply a physically consistent lag adjustment that aligns Kratie discharge with the confluence response.*

**5. L126: The Delft3D-Flow hydrodynamic model first appeared in Section 2.2. It would be better to mention it first in the Introduction.**

**Response:** Thank you for the helpful suggestion. Please refer to comment 4.

**6. L133: (see Figure 2d–f) should be moved to the end of the sentence.**

**Response:** Thanks for this suggestion. We move (see Figure 2d–f) to the end of the sentence, and read as follows:

*Except for this case, it is important to emphasize that our analyses reflect the compounded impacts of dam regulation and climate variability, without explicitly disentangling their relative contributions (see Figure 2d–f).*

**7. L140: Provide the full form of PMFM.**

**Response: Thanks for your comment. We consider this comment and the revise sentence reads as follows:**

*For hydrodynamic analyses of the Tonle Sap system, additional time series of lake water levels, reverse flow periods, and discharge were acquired from the MRC and Procedures for the Maintenance of Flows on the Mainstream (PMFM) online platform (https://pmfm.mrcmekong.org/monitoring/6b/)*

**8. L180: Should (see Sect. 2.5.2) actually refer to Sect. 2.5.3? Please also verify all other cross-references.**

**Response:** Thanks for the comment, and apologies for the oversight. Yes, it refers to Section 2.5.3.

**9. L192: Figure 1c should be Figure 1b. Please correct.**

**Response:** Thanks for the comment. Yes, it should be Figure 1b. The revise text reads as follows:

*To represent these flows, we extracted canal networks using a machine learning–based remote sensing model developed by Zhao et al. (2025), supplemented by manual digitization from high-resolution satellite imagery (see Figure 1b).*

**10. L239: Please mention the period of calibration and validation for the model in the main text, although this appears in the supplementary file.**

**Response:** Thanks for the comment. We revise the text as follows:

*Model calibration was conducted using an automatic parallel computation framework that optimizes hydrological parameters across multiple REWs simultaneously (Nan et al., 2021) (Table S1). The THREW model was calibrated for 2000–2009 and validated for 1980–1999 using observed discharge at available gauging stations; detailed performance metrics are provided in the Supplement.*

**11. L279: Change (a-f) to (Figure 2a-f) or (Figure 2)**

**Response:** Thanks for the comment. We revise the text, which reads as follows:

*Six rose diagrams (Figure 2a-f) summarize the spatiotemporal evolution of discharge dynamics across eight mainstream stations from Chiang Saen to Kratie.*

**12. Figure 4: The cross-reference of Figure 4d is apparently missing in the text.**

**Response.** Thank you for noting this omission. We revise the Results text to explicitly cite Fig. 4d, which shows the timing of annual minimum discharge across stations.

*The timing of annual minimum flows has consistently advanced across all stations along the Mekong mainstream (Figure 4d). At Kratie, for example, the timing of median minimum discharge shifted from early April (April 10)*

**13. L408: Figure 6a should be 5a.**

**Response.** Thank you for pointing out this error. We correct the cross-reference at L408; "Fig. 6a" is changed to "Fig. 5a", as follows:

*Figure 5a highlights two critical elements: (i) the nonlinear discharge–travel time relationship between Kratie and the confluence, and (ii) …….*

**14. L421 and L431: Panel 5b and Panel 5c should be addressed as Figure 5b and Figure 5c.**

**Response:** Thank you for noting this. We revise the text at L421 and L431 to replace "Panel 5b/5c" with the correct figure references, "Fig. 5b" and "Fig. 5c," respectively

**15. Figure 5a: It is hard to identify the monthly average discharge during the post-dam period. Please improve the figure.**

**Response:** Thank you for the comment. We have revised Fig. 5a to improve readability, particularly for distinguishing the monthly average discharge during the post-dam period (see below).

---

## Author Comment (AC2)

The authors thank the reviewer for the positive comments, constructive feedback, and helpful suggestions. Below, we provide point-by-point responses outlining how each comment are addressed in the revised manuscript. Our replies are introduced by "Response:". Text highlighted in blue indicates additions or revisions proposed for inclusion in the updated manuscript.
* * *
**General comment:**

**The study addresses an important and timely problem by examining how hydropower development and climate change have altered the flow regime of the Mekong River. Using historical observations and a combination of hydrological and hydrodynamic modelling, the authors estimate statistical indicators of flow memory and synchrony. They show substantial post-dam changes at several mainstem gauges when compared to their pre-dam equivalents. This component of the analysis is methodologically sound, well supported by the results presented and broadly consistent with previous studies that have shown dampened wet-season flow peaks and enhanced dry-season flows under dam regulation. The attempt to link these changes in Mekong flow discharge to alterations in the reversal of the Tonle Sap River however is less convincing. The supporting results are limited relative to the strength of the claims and key relevant literature on the Tonle Sap-Mekong system is not adequately considered. As a result, the conclusions regarding reverse-flow dynamics would require more rigorous modelling and analysis to improve the manuscript. Please find my specific comments and technical corrections below:**

**Response:** Thank you for your feedback. Below, we summarize how we will revise the manuscript to address your comment.

**Specific comments:**

**The terms 'flow memory' and 'flow synchrony' are used throughout the manuscript including the title, but they are not fully introduced and their significance is not properly explained. I suggest that a short description is added in the introduction.**

**Response:** We agree. We add a short description defining flow memory as long-range persistence in discharge variability (quantified using the Hurst exponent) and flow synchrony as the coherence of hydrograph timing/variability across mainstream stations and clarify why both properties are central to diagnosing basin-wide reorganization of the flood pulse and downstream river–lake responses. These edits are added in the Introduction, and linked to the corresponding Methods description. The revised section reads as follows:

*"While such studies have advanced our understanding of hydrological alterations, limited empirical research has examined how regulation disrupts intra-annual flow memory—eroding the natural seasonal recurrence and persistence of discharge fluctuations—or fragments flow synchrony across stations, thereby decoupling linked hydrological and ecological responses (Poff et al., 2007). Here, flow memory refers to the persistence (long-range dependence) in discharge variability, i.e., the extent to which present conditions retain information from antecedent fluctuations; we quantify this property using the Hurst exponent (Methods). Flow synchrony refers to the degree of spatiotemporal coherence among mainstream hydrographs—how consistently discharge rises, peaks, and recedes across stations—and is evaluated using inter-station similarity of standardized daily discharge time series (Methods). Disrupted memory and fragmented synchrony therefore capture not only shifts in flow magnitude, but also a loss of basin-scale coherence that can weaken the propagation of flood-pulse signals and reduce the predictability of hydrologic cues relevant to downstream processes.*

**Lines 21-23: The statement made here is misguided. The discharge threshold that must be exceeded in order for the Tonle Sap River to reverse its flow is governed by the geometrical characteristics of the Tonle Sap and Mekong Rivers and their confluence. These geometrical characteristics are indeed altered by sand-mining induced channel deepening. Mekong's flow discharge, which is affected by dam regulation and climate change, may control whether and for how long this threshold is exceeded but does not affect the threshold itself as implied here.**

**Response:** Thank you for this important clarification. We agree that the **instantaneous hydraulic condition for Tonle Sap flow reversal** is governed primarily by the **geometry and conveyance of the Mekong–Tonle Sap confluence and the associated stage–discharge relationships**, which can be modified by sand-mining–driven channel incision. In our manuscript, the "discharge threshold" refers to the **lag-adjusted Kratie discharge at which reverse flow is observed to initiate/cease** (Fig. 5), i.e., an **operational proxy** for the confluence-stage condition rather than a fixed geometric constant. We revise the Abstract and Conclusion to clarify that **riverbed lowering increases the discharge required to reach the confluence stage needed for reversal (via a shifted stage–discharge relation)**, whereas **dam regulation and climate change primarily influence how often and for how long this condition is exceeded**, thereby shortening the reverse-flow season. We also add a brief clarification in Section 3.5 defining this usage of "threshold."

**Abstract:**

*"Critically, observed riverbed lowering from sand mining has likely shifted the local stage–discharge relationship near the Phnom Penh–confluence reach, such that a higher Kratie discharge is now required to attain the confluence stage associated with reverse-flow initiation: the median onset discharge increased from ~3,000 m³ s⁻¹ (pre-dam) to ~7,000 m³ s⁻¹ (post-dam), an increase of >130%. Dam regulation and climate change mainly modulate whether and for how*

*long this hydraulic condition is exceeded, contributing to a 24-day shortening of the reverse-flow season relative to the historical baseline.*

**Section 3.5**

*"Here, "threshold" denotes the lag-adjusted Kratie discharge associated with the observed onset/cessation of reverse flow, used as an operationally interpretable proxy for the hydraulic condition at the confluence. It is therefore influenced by local stage–discharge relations (including incision effects) and by the timing-dependent head difference between the mainstream and the lake, rather than representing a fixed geometric constant."*

**Conclusion**

These mainstream alterations have propagated into the Tonle Sap system. The reverse-flow period has shortened by approximately 24 days during the post-dam period compared to the pre-dam baseline. Importantly, the hydraulic threshold for reversal at the Mekong–Tonle Sap confluence is governed primarily by local channel geometry and confluence hydraulics (i.e., the stage/head difference required for flow reversal). Within this framework, sand-mining–driven riverbed incision lowers confluence stage for a given discharge, thereby increasing the lag-adjusted Kratie discharge required to reach the confluence stage associated with reversal. Consistent with this mechanism, the median lag-adjusted Kratie discharge associated with reverse-flow initiation increased from ~3,000 $m^3\ s^{-1}$ (pre-dam) to ~7,000 $m^3\ s^{-1}$ (post-dam), representing a >130% increase. In contrast, dam regulation and climate variability primarily control whether—and for how long—this (now higher) discharge is exceeded, reducing exceedance frequency and persistence and thereby shortening and retiming the reverse-flow season.

**Lines 40-50: The authors correctly argue that previous literature has already extensively studied the impacts of hydropower development on Mekong hydrology. Here also please specify the start and end of for the dry and wet seasons.**

**Response:** Thanks for your suggestion. We revise the text as follows:

These impacts have been especially pronounced during the dry season (November–April): Räsänen et al. (2017) reported dry season discharge increases of 121-187% at Chiang Saen, the most upstream station in our study reach, in March, and an increase of 32-46% at Kratie, the most downstream station (Fig. 1). Lu and Chua (2021) found a 98% increase in monthly discharge at Chiang Saen during the dry months. Concurrently, wet-season flows (May–October) have declined substantially (Lu et al., 2014), undermining the amplitude and timing of flood pulses that sustain floodplain ecosystems. Nguyen et al. (2025) documented a 73.7% increase in dry-season flows at Chiang Saen between 2000 and 2019, underscoring the dominant role of dam-induced regulation.

**Lines 51-53: The authors here argue that the novelty of their study is on the incorporation of flow memory and synchrony but they do not explain this further or justify why this analysis is important.**

**Response:** Thank you for this constructive comment. We agree that, in the previous version, the manuscript stated the novelty of incorporating *flow memory* and *flow synchrony* without sufficiently explaining **why these properties matter physically and ecologically**, and how they advance beyond conventional station-based alteration metrics. We therefore **expand the Introduction** at the first mention of these concepts to (i) clarify that memory and synchrony quantify **temporal persistence** and **network-scale coherence** of the hydrograph, respectively, and (ii) explain why their disruption is important for **flood-pulse integrity** and for **threshold-sensitive river–lake exchange**, including Tonle Sap reverse flow. These additions explicitly connect altered memory/synchrony to the persistence of the hydraulic gradient and to the predictability and propagation of seasonal flow cues. We also **strengthen the final paragraph of the Introduction** to more clearly articulate the study's novelty: the joint diagnosis of long-term memory, inter-station synchrony, and sub-daily variability across eight stations (1976–2024), combined with a hydrodynamic response-time framework to link these multi-scale alterations to long-term shifts in reverse-flow onset/cessation thresholds.

**Lines 53-66: Here the focus shifts towards the Tonle Sap River flow reversal but key literature that has studied this topic is omitted. The authors should review previous work focusing on the effects of climate change and flow modulation by dams (see for example: Wang et al., Environ. Res. Lett. 15, 0940a1 (2020); Frappart, F. et al Sci. Total Environ. 636, 1520–1533 (2018); Kummu and Sarkkula, Ambio 37, 185–192 (2008)) and sand mining (see for example: Quan L.Q. et al., Nat Sustain 8, 1455–1466 (2025)). When relevant literature is considered, the statement made in lines 63-66 is not supported.**

**Response:** Thank you for highlighting these key references. We incorporate them into the introduction.

**Line 131: Please specify the quality checks that were applied to the data, what proportion of the data did not pass the quality control and how were these values replaced.**

**Response:** Thanks for raising this point. For almost all stations, reliable and continuous daily records are available; however, in the Mekong River Commission dataset we occasionally found isolated missing days. For these gaps, we interpolated daily values using the adjacent observations (the preceding and following days). We also provide detailed information on data quality control, including the proportion of records that did not pass the quality checks.

**Line 171: Figure 1 does not show the Delft 3D model domain, I suggest making a separate figure for this.**

**Response:** Thanks for your comment. We draw a new Figure and include Delft3D flow domain. Thanks

**Lines 195-196: Please specify the distance between neighbouring cross-sections. This could improve confidence in the bathymetric interpolation that was applied.**

**Response:** Thanks for your suggestion. Our bathymetric dataset covers the Mekong Delta, Tonle Sap River, Mekong River, and Tonle Sap Lake, with surveyed cross-sections spaced approximately 300–3,000 m apart. We provide a detailed description of the interpolation workflow used to transform these cross-sections into a continuous, grid-based bathymetric DEM for the model computational domain.

**Lines 200-203: Could you show how well does the derived DEM approximate natural river and lake morphology. Can the error be quantified? It should be noted here that the supplement provides only a single cross-section as an example of good fit (Figure S1).**

**Response:** Thanks for your comment. We quantified interpolation performance using leave-one-section-out cross-validation and comparisons against independent measured transects. This evaluation yielded an RMSE of 0.46 m and a median absolute error of 0.37 m. we add these details into the revised manuscript.

**Lines 234-238: Please provide more detailed information on the data that were used in the hydrological model, including sources.**

**Response:** Thank you for this comment. We agree that the meteorological forcing and other THREW input datasets should be explicitly documented. We revise Section 2.3 to provide the variables and sources used to force THREW (precipitation, temperature, and Penman–Monteith potential evapotranspiration), as well as the land-surface/vegetation datasets used for parameterization (soil properties and MODIS-based vegetation/snow products):

In hydrological model section, we add the following details:

*The THREW hydrological simulations were driven by station-based precipitation and meteorological observations from the Mekong River Commission (MRC) and the China Meteorological Administration (CMA). Precipitation was obtained from a basin-wide gauge network (105 stations) and air temperature from 35 stations (Fig. S3). Daily potential evapotranspiration was computed using the Penman–Monteith method based on station meteorological variables (including temperature, wind speed, humidity, and radiation/sunshine duration; Fig. S3). Soil properties were taken from the FAO global soil database (10 km).*

*Vegetation and surface-condition inputs (NDVI, LAI, and snow cover) were derived from MODIS products (500 m, 16-day) following Zhang et al. (2023).*

**Lines 276-277: The manuscript here refers to the supplement for detailed validation of the models used in the study. Figure S5 of the supplement shows that on some occasions the model underpredicts peak discharge values for Kompong Luong while simultaneously overpredicting for Prek Kdam (the opposite also occurs), what are the implications of these discrepancies for the simulated hydraulic head and the reversal of the Tonle Sap River? Can you clarify what is an 'acceptable limit' for RMSE values, mentioned in the supplement right before Figure S5?**

**Response:** Thanks for your comment. We understand that the reviewer is likely referring to Fig. S5, which compares simulated and observed water levels at two key stations (**rather than discharge**). There is no universal "acceptable" RMSE threshold, as it depends on model purpose, domain complexity, boundary/forcing uncertainty, and observation quality. In our case, we used high-quality input datasets and simulated a large, hydraulically complex river–lake–floodplain system; within this context, model performance is strong. Across the evaluation years, water-level errors correspond to an RMSE of ~12% relative to the observed variability. Importantly, the variables most critical to our study—exchange flow between Tonle Sap Lake and the Mekong mainstream and the timing of reverse-flow onset and cessation—are reproduced with high fidelity (Fig. S6; Table S2). We add these clarifications to the Supplementary Information.

**Lines 279-282: A sentence should be added here to explain that the no-dam scenario data presented are outputs from the THREW model.**

**Response:** Thanks for your suggestion. We will add this sentence.

**Lines 292-300: The RBI patterns described in the text do not reflect what is shown in Figure 2 panels b and e. In addition to the discrepancies between text and Figure 2, post-dam RBI (panel b) and measured RBI (panel e) should be identical, but this is not the case here.**

**Response.** The comment is valid, as panel e was not fully consistent with panel b, particularly in how the post-dam period pattern was presented. We appreciate the reviewer for bringing this to our attention, and we redraw the figure during the revision process to ensure consistency across panels.

**Lines 413-415: It is unclear what these lines refer to. Earlier it has been demonstrated (Lines 355-377) that the median value of the annual maximum discharge is reduced in the post-dam period by 9%. RBI flashiness is also reduced at Kratie in the post-dam period based on Figure 2. These results do not support the claim of intensification of the hydropeaks.**

**Response: Response:** Thank you for this comment. We agree that the original wording in Lines 413–415 was **unclear and overly general**, and could be interpreted as implying an intensification

of hydropeaking at **Kratie**, which is not supported by our results. As shown in Lines 355–377, the median annual maximum discharge at Kratie decreases in the post-dam period (~9% reduction), and sub-daily flashiness is attenuated downstream (Fig. 2–3). Our intention was to highlight that **sub-daily hydropeaking is spatially heterogeneous** and can remain pronounced at upstream–midstream stations even if its expression is damped at Kratie, and that **short response times (1–3 days in wet season)** make robust travel-time estimates important for operational preparedness. We have therefore revised Lines 413–415 to explicitly reflect the downstream attenuation at Kratie while retaining the motivation for travel-time analysis. We aim to revise the text as follows:

*"Although daily variability and annual peak discharge at Kratie are dampened in the post-dam period (Figs. 2 and 4), sub-daily hydropeaking remains pronounced at several upstream–midstream stations (Fig. 3) and can propagate downstream. Given the short wet-season response time (1–3 days), robust travel-time estimates remain essential for short-lead warning and operational preparedness in Phnom Penh and the delta."*

**Lines 421-425: The argument here is constructed in a confusing way and it is not clear why cessation of the reversal of the Tonle Sap River requires high discharge values at Kratie.**

**Response: Response:** Thank you for noting this ambiguity. We agree the original wording was confusing and could be interpreted as implying that high discharge is required *to cause* cessation. In our analysis, the "cessation threshold" denotes the **lag-adjusted Kratie discharge at the time reverse flow ends** (i.e., when the Mekong–lake hydraulic gradient is no longer positive). Because cessation occurs on the **falling limb** while Tonle Sap Lake levels remain elevated from seasonal storage, the discharge at cessation can still be relatively high. We revised the text in Section 3.5 to state explicitly that reverse flow **ceases when Kratie discharge falls below a sustaining level**, and we added a brief explanation of this onset–cessation hysteresis.

We aim to revise the text as follows:

*Figure 5b presents the lag-adjusted Kratie discharge associated with the **onset** (rising limb) and **cessation** (recession limb) of reverse flow into Tonle Sap Lake. In the pre-dam period (1976–1991), reverse flow typically initiated when discharge exceeded ~3,000 m³ s⁻¹ and **ceased when discharge fell to ~28,000 m³ s⁻¹**. In the post-dam period (2010–2024), the onset discharge increased to ~7,000 m³ s⁻¹, while reverse flow **ceased when discharge declined to ~34,000 m³ s⁻¹** (with some years near 40,000 m³ s⁻¹). The higher cessation value reflects **onset–cessation hysteresis**: onset occurs early on the rising limb when lake levels are low, whereas cessation occurs later during drawdown when the lake remains elevated, so sustaining a positive Mekong-to-lake head difference requires comparatively larger discharge/stage.*

**Lines 427-428: The authors omit the study published by Quan et al., Nat Sustain 8, 1455–1466 (2025) which demonstrates the impacts of sand mining on the flow reversal of the Tonle Sap River.**

**Response:** Thanks for your comment. At the time of our initial submission, the study by Quan et al. had not yet been published. We are aware of this important work and now cite it in several relevant sections of the revised manuscript.

**Lines 435-436: The effect of channel deepening which has drastically changed the hydraulic head required to reverse the flow of the Tonle Sap River should be included here**

**Response:** Thanks for your suggestion. We agree that addressing this point will strengthen the discussion in this section, and we incorporate it into the revised manuscript.

**Lines 439-444: The argument here is speculative. The figure shows very well the modulation of Mekong's water flux in the post-dam era and the shortening of the duration of the TSR reversal but do not show the drivers for these changes.**

**Response:** We agree with the reviewer. The previous wording overstated attribution of the patterns in Fig. 5 to specific drivers. Fig. 5 is intended to describe changes in hydrograph shape, lag-adjusted onset/cessation thresholds, and reverse-flow timing; it does not isolate causal contributions of dam regulation, climate variability, or sand mining. We therefore revise the text to remove causal language ("drivers are implicated", "primarily driven", "shaped by…") and replaced it with wording that (i) reports the observed shifts and (ii) clarifies that mechanistic attribution is discussed separately and remains beyond the scope of Fig. 5.

**Lines 484-485 The statement here is misguided. The reduction of Mekong water discharge cannot affect the threshold required to initiate flow reversal in the Tonle Sap River. This threshold is governed by channel geometry. The magnitude of the hydropeak affects when and for how long this threshold is exceeded and reversal occurs.**

**Response:** The comment is valid. Please refer to our replies to previous comment.

**Figure 1: The labels of the panels need attention as two panels are labelled as '(a)'. Then later in the main text (line 192) 'Figure 1 panel c' is mentioned. Also, you should provide the sources for the data presented (for example on dam locations).**

**Response:** The two panels labelled "a" in Fig. 1 were intended to indicate that they are directly linked: the larger panel provides a zoomed-in version with additional detail of the smaller panel. We acknowledge the lettering inconsistency (e.g., the use of "c") and revise it accordingly. Thank you.

**Figure 2: I am not convinced that the use of rose diagrams is appropriate here. For example, a connection between the furthest upstream station (Chiang Saen) and most downstream (Kratie) is implied. I suggest using line plots with stations placed in order along the x-axis, possibly with**

**the in-between distances scaled according to the station km point along the Mekong mainstem. Also, please explain the abbreviations used for the names of the stations in the figure caption.**

**Response:** Thank you for the suggestion. We agree that a rose diagram can be misread as implying a circular connection between the most upstream and most downstream stations. Our intention, however, is not to represent geographic connectivity but to provide a **compact comparative display** of multi-station changes across hydrological periods/metrics, where the circular layout improves readability relative to multiple line panels with many overlapping series. To avoid misinterpretation, we have revised Figure 2 by (i) explicitly ordering stations clockwise from upstream to downstream and adding a clear "Upstream → Downstream" directional annotation, (ii) visually separating the first and last stations (e.g., a gap/break marker) to prevent any implied closure, and (iii) adding an inset longitudinal line plot with stations placed along the x-axis (scaled by river-km where available) to directly convey the upstream–downstream gradient. We also expanded the figure caption to define all station abbreviations.

**Figure 3: Presenting data as monthly averages using all years (right panels) and annual averages (left panes) suppresses information that would be helpful to understand spatio-temporal changes of the metrics. For example, the monthly data show an uptick for Nakhon Phanom in amplitude and flashiness for January, is this primarily driven by the huge spike in 2022?**

**Response:** Thank you for this helpful point. We agree that pooling all years into monthly means (right panels of Fig. 3) can mask interannual variability and potentially allow a single anomalous year to disproportionately affect the monthly climatology. Because sub-daily water-level records are only available from 2018–2024 and because the monthly metrics are currently summarized as means, we revise the presentation to make year-to-year variability explicit. Specifically, we adde a new figure showing year-resolved monthly amplitude and flashiness for Nakhon Phanom (and key stations), allowing direct evaluation of whether the elevated January values are driven primarily by 2022 or represent a more persistent pattern. We also clarify in the Fig. 3 caption and Sect. 3.3 that the right panels show pooled multi-year monthly means.

**Also on Figure 3: Extreme values in 2018 for Chiang Khan (amplitude and flashiness) and Pakse (all metrics) and 2022 Nakhon Phanom (all metrics) should be discussed in the text.**

**Response:** We agree and revise the Results text to explicitly discuss the prominent interannual extremes evident in Fig. 3. We clarify that these "extreme" values represent years with unusually strong **sub-daily stage variability and/or frequent rapid ramping events** (peak count ≥ 5 cm hr$^{-1}$), rather than implying increases in annual peak discharge. Thanks for your comment.

**Technical corrections:**

**Line10: add 'the' before Tonle Sap Lake**

**Response:** The comment is considered. Thanks

**Line 99 and throughout the manuscript MCM is not a universal abbreviation. I suggest to use M m3**

**Response:** We agree. To avoid ambiguity, we replaced "MCM" with using "×10$^6$ m$^3$" (million cubic metres) and updated the first occurrence accordingly.

**Line 126: replace 'part (a)' with 'panel (a)'**

**Response.** Thanks for your suggestion. We revise the sentence.

**Line 170: delete 'of the Sea'**

**Response:** Thanks for suggestion. We delete "of the Sea"

**Line 180: the hydrological model is described in Sect. 2.5.3**

**Response:** Thanks for your comment. Following the reviewer's suggestion (reviewer 1), we now present the hydrological model first as Section 2.5.1. We have revised the text accordingly to ensure consistency between the section numbering and the in-text references.

**Line 408: the correct figure is 5a (not 6a)**

**Response.** We apologize for the oversight. We change"6a" to "5a" as suggested.

**Line 421: Replace 'Panel 5b' with 'Figure 5b'**

**Response.** We apologize for the oversight. We replace 'Panel 5b' with 'Figure 5b' as suggested.

**Throughout the manuscript there is an overuse of em dashes (–). In most cases these should be replaced with commas or with hyphens when used for ranges (e.g. 1976-1991, not 1976–1991) or connections (e.g. Tonle Sap-Mekong , not Tonle Sap–Mekong).**

**Response:** We agree that dashes were overused. We revised the manuscript to reduce em-dash usage by replacing many instances with commas, parentheses, or sentence restructuring. We also standardized punctuation consistently: hyphens are used for compound modifiers (e.g., post-dam, lag-adjusted), while en dashes are retained for numeric ranges (e.g., 1976–1991) and named linkages (e.g., Mekong–Tonle Sap), following common journal style.